# Hematuria Is Associated with More Severe Acute Tubulointerstitial Nephritis

**DOI:** 10.3390/jcm9072135

**Published:** 2020-07-07

**Authors:** Raquel Esteras, Jonathan G. Fox, Colin C. Geddes, Bruce Mackinnon, Alberto Ortiz, Juan Antonio Moreno

**Affiliations:** 1Renal Laboratory, Health Research Institute-Fundación Jimenez Diaz (IIS-FJD), Autónoma University of Madrid (UAM), 28040 Madrid, Spain; raquel.esteras@quironsalud.es; 2Glasgow Renal and Transplant Unit, Queen Elizabeth University Hospital, Glasgow G51 4TF, UK; foxjg@aol.com (J.G.F.); colin.geddes@ggc.scot.nhs.uk (C.C.G.); 3Department of Nephrology and Transplantation, John Hunter Hospital, Newcastle, NSW 2310, Australia; Bruce.Mackinnon2@ggc.scot.nhs.uk; 4Department of Cell Biology, Physiology and Immunology, University of Cordoba, 14041 Cordoba, Spain; 5Maimonides Biomedical Research Institute of Cordoba (IMIBIC), Reina Sofia University Hospital, 14004 Cordoba, Spain; 6Centre of Biomedical Research in network of Cardiovascular disease (CIBERCV), 28029 Madrid, Spain

**Keywords:** acute kidney injury, acute tubulointerstitial nephritis, haematuria, kidney biopsy, outcomes, proteinuria, chronic kidney disease

## Abstract

Acute tubulointerstitial nephritis (ATIN) is a common cause of acute kidney injury. Although haematuria is a risk factor for the development of renal disease, no previous study has analyzed the significance of haematuria in ATIN. Retrospective, observational analysis of 110 patients with biopsy-proven ATIN was conducted. Results: Haematuria was present in 66 (60%) ATIN patients. A higher percentage of ATIN patients with haematuria had proteinuria than patients without haematuria (89.4% vs. 59.1%, *p* = 0.001) with significantly higher levels of proteinuria (median (interquartile range) protein:creatinine ratio 902.70 (513–1492) vs. 341.00 (177–734) mg/g, *p* <0.001). Moreover, those patients with more haematuria intensity had a higher urinary protein:creatinine ratio (1352.65 (665–2292) vs. 849.60 (562–1155) mg/g, *p* = 0.02). Those patients with higher proteinuria were more likely to need renal replacement therapy (22.7 vs. 0%, *p* = 0.03) and to suffer relapse (4 vs. 0%, *p* = 0.03). At the end of follow up, haematuric ATIN patients had higher serum creatinine levels (3.19 ± 2.91 vs. 1.91 ± 1.17 mg/dL, *p* = 0.007), and a trend towards a higher need for acute dialysis (7 vs. 1%, *p* = 0.09) and renal replacement therapy (12.1 vs. 2.3%, *p* = 0.12). Haematuria is common in ATIN and it is associated with worse renal function outcomes.

## 1. Introduction

Acute tubulointerstitial nephritis (ATIN) is a common cause of acute kidney injury (AKI) [1], representing 15–25% of cases [2,3] where the indication for renal biopsy was AKI. However, it is less common than other causes of AKI such as hypoperfusion or sepsis [4].

Although the precise incidence of ATIN is not known, recent studies suggest that it is increasing, especially in elderly patients [5]. The clinical presentation may include fever (36%), skin rash (22%) and eosinophilia (35%) [5,6], but the simultaneous presence of these signs is observed in less than 10–15% of cases [7]. Other frequent clinical manifestations are non-nephrotic proteinuria, leucocyturia and haematuria [5,6].

Haematuria may be a manifestation of many renal diseases as well as ATIN. In the past, the occurrence of haematuria was thought to be innocuous for the kidney but recent reports have propelled haematuria to the forefront of clinical nephrology. Thus, the presence of isolated haematuria in young individuals is associated with an increased risk of end-stage renal disease (ESRD) [8]. Moreover, haematuria has been found to predict renal outcome in IgA nephropathy, being associated with an increased risk of ESRD after 10 years of follow-up [9]. Disappearance of both haematuria and proteinuria has recently been proposed to define clinical remission in IgA nephropathy [10,11]. Along the same lines, we recently observed that up to 25% of IgA nephropathy patients with haematuria-associated AKI do not recover baseline renal function, suggesting long term negative effects [12]. The deleterious effects of haematuria include direct tubular damage by intratubular obstruction by red blood cell casts, toxic effects of haemoglobin (Hb) and haem on kidney epithelial cells, and erythrophagocytosis by renal tubular cells [13]. In tubular cells, Hb induces oxidative stress, cell death, and pro-inflammatory and pro-fibrotic responses [13].

Few studies have analysed the presence of haematuria in ATIN and, therefore, the precise incidence of haematuria is not well known. Some smaller reports (21–60 patients) have shown haematuria to be present in 68–95% of ATIN patients [1,14,15,16], while a larger study (*n* = 130) found haematuria in 30% of patients [16]. However, no previous reports have analysed the association of haematuria with renal outcomes in ATIN. Thus, the aim of the present study was to assess the prevalence of haematuria in a large cohort of ATIN patients and to report, for the first time, whether haematuria is associated with kidney outcomes.

## 2. Experimental Section

### 2.1. Patients

We performed a retrospective, observational study of biopsy-proven ATIN cases between January 2010 and December 2018 using data available in the Glasgow Renal Biopsy Registry. All subjects gave their informed consent for inclusion before they participated in the study. The Registry was conducted in accordance with the Declaration of Helsinki and with institutional ethics committee approval, which covered this study. It is the registry of the Glasgow Renal and Transplant Unit, a large tertiary adult nephrology centre which serves a population of 1.5 million in the west of Scotland. A total of 174 patients had biopsy-proven ATIN. Specimens were examined by a specialist renal pathologist to confirm the diagnosis of ATIN; cases with another primary renal diagnosis and co-existing interstitial nephritis were excluded. Patients without dipstick data at ATIN diagnosis were also excluded from the analysis (Figure 1), leaving 119 patients with haematuria information. An additional 9 patients were excluded because of the coexistence of a pathological diagnosis of ATIN with a second pathological diagnosis of a known cause of haematuria such as glomerulonephritis or vasculitis. This left 110 patients for the main analysis of potential differences between patients with and without haematuria with respect to age, sex, comorbidities, renal function, aetiology and outcomes (development of CKD, acute/chronic dialysis, kidney transplant, relapse). An analysis of histological findings in kidney biopsies was also performed on the 104 samples in which these were available (Figure 1).

A review of case notes and the electronic laboratory data archive was undertaken to determine patient demographics and presenting clinical features (including fever, rash and eosinophilia—defined as a count above the laboratory reference range during AKI), as well as physician-defined aetiological factors. Identification of a drug-related aetiology required the prescription of a known causative drug in the absence of systemic diseases associated with ATIN or untreated infections. In cases of multiple potential causative drugs, the likely agent was identified on the basis of temporal relationship to the AKI.

Available renal function data from before ATIN diagnosis (baseline), at ATIN presentation (considering the first evidence of AKI as the onset), at the peak of AKI (renal function nadir), at the time of renal biopsy and at the end of follow-up were analysed. Patients were followed until complete kidney function recovery, end of the treatment with corticosteroids, death or loss of tracking due to lack of data in the record system. Recovery of renal function was defined as an end of follow-up serum creatinine within ±0.3 mg/dL of baseline serum creatinine. Proteinuria was defined as ≥150 mg/g of urinary creatinine or ≥250 mg in 24 h and albuminuria as ≥30 mg/g of urinary creatinine. Eosinophilia was defined as the presence of ≥0.5 × 10^3^ eosinophils per µL of blood.

The diagnosis of ATIN was based on the kidney biopsy histology i.e., the presence of interstitial inflammatory infiltrates composed of lymphocytes, monocytes, eosinophils, plasma cells and neutrophils in some cases, accompanied by different degrees of acute tubular damage and oedema. Haematuria data were obtained from dipstick urinalysis and classified according to severity (+ mild, ++ moderate, +++ severe). The median (interquartile range, IQR) time from onset serum creatinine and performance of a renal biopsy was 57 (18–144) days.

### 2.2. Statistical Analysis

Quantitative variables with a normal distribution are presented as mean ± standard deviation (SD) and were compared using the Student *t* test. Data that did not follow a normal distribution are displayed as median (interquartile range, IQR) and compared with the Mann–Whitney test. Qualitative variables are shown as percentages and compared with the Chi-square or the Fisher’s test when appropriate. Variables with a *p* < 0.05 were considered statistically significant. Statistical analysis was performed using SPSS 19.0 statistical software (SPPS, Inc, Chicago, IL, USA).

## 3. Results

Among the 174 ATIN patients initially identified, 119 (68.4%) had urinary dipstick data; of these, 110 patients had a pathological diagnosis of AKI alone and were included in the analysis. As reported in Table 1, 66 patients (60%) had haematuria at diagnosis of AKI whereas 44 did not. No significant differences in age, sex, comorbidities or baseline renal function were observed between haematuric and non-haematuric patients.

There were no statistically significant differences between haematuric and non-haematuric patients in onset, time of biopsy or peak serum creatinine (Table 2). However, at time of biopsy, patients with haematuria had a higher urinary protein:creatinine ratio (UPCR) (902.70 (513.3–1492) vs. 341.00 (177–734) mg/g, *p* < 0.001) (Table 2). At this time-point, a higher percentage of ATIN patients with haematuria had proteinuria than patients without haematuria (89.4 vs. 59.1%, *p* = 0.001) (Table 2). There were no differences in other blood parameters, including eosinophilia or CRP, or in clinical manifestations (fever, skin rash, hypertension or development of nephrotic syndrome). At the end of follow-up, ATIN patients with haematuria had higher serum creatinine levels (3.19 ± 2.91 vs. 1.91 ± 1.17 mg/dL, *p* = 0.007). There was a non-significant trend towards a higher need for acute dialysis (7 vs. 1%, *p* = 0.09) and chronic renal replacement therapy at the end of follow-up in haematuric rather than in non-haematuric ATIN patients (12.1 vs. 2.3%; *p* = 0.12) (Table 3).

There were no differences in other blood parameters, including eosinophilia or CRP, or in clinical manifestations (fever, skin rash, hypertension or development of nephrotic syndrome) (Table 2), or cause of ATIN (Table 4) between haematuric and non-haematuric ATIN patients. However, it is observed that ATIN caused by non-steroidal anti-inflammatory drugs (NSAIDs) is more frequent in patients with haematuria (13.6 vs. 4.5%. *p* = 0.08). Regarding histological features, interstitial inflammatory cell infiltration was the most common finding, frequently occupying more than 50% of the tissue section in both groups. Eosinophils, lymphoplasmacytic cells and macrophages were the most common cells observed in renal biopsies. The only significant difference between groups was the more frequent presence of red blood cell casts (10.6 vs. 2.3%, *p* = 0.006) in patients with haematuria (Table 5), which was in line with urinary findings.

Stratification of patients according to severity of haematuria reported that those patients with more haematuria intensity had a higher urinary protein:creatinine ratio (UPCR) (1352.65 (665–2292) vs. 849.60 (562–1155) mg/g, *p* = 0.02). Additionally, these patients also had higher C reactive protein (CRP) (77.51 ± 91.07 vs. 41.25 ± 48.91 µg/L, *p* = 0.05) and a lower number of eosinophils in their blood (0.26 ± 0.16 vs. 0.39 ± 0.23 *p* = 0.04) (Table 6).

Most patients (*n* = 90/110, 81.8%) were treated with steroids, usually after renal biopsy. There were no differences in haematuria prevalence between patients treated or not treated with steroids (55/90, 61.1% vs. 11/20, 55%, *p* = 0.62). Moreover, there were no significant differences in initial serum creatinine or disease course between patients treated or not treated with steroids, although the percentage of patients who recovered their previous renal function was higher in the steroid group (41.1 vs. 15%, *p* = 0.12) (Appendix A).

According to proteinuria levels, those patients with higher proteinuria showed a significantly higher need of chronic renal replacement therapy (22.7 vs. 0%, *p* = 0.03) and relapse (4 vs. 0%, *p* = 0.03) (Appendix A).

## 4. Discussion

Although haematuria is a common finding in patients with ATIN, our results suggest that clinicians pay little attention to haematuria in this condition as haematuria data were recorded in 60% of ATIN patients. Importantly, we found haematuria in biopsy-proven ATIN to be associated with proteinuria during the AKI episode and worse renal function outcomes.

Haematuria may cause tubular cell injury contributing to AKI and CKD [8,9,12]. Epidemiological studies have associated the presence of haematuria with an increased risk of developing ESRD. In Israeli youths, the presence of isolated microhaematuria was associated with an increased risk of ESRD after 22 years of follow-up [8]. Other studies have shown an association of microhaematuria with faster CKD progression [17]. Findings from the Chronic Renal Insufficiency Cohort (CRIC) study and the Evaluating Prevention of Progression in Chronic Kidney Disease (EPICC) trial, suggest an increased risk or ESRD after two years of follow-up in patients with baseline haematuria [18]. We now present the first study analysing the association of haematuria with disease severity and outcomes in a large population of biopsy-proven ATIN patients. In our study, haematuria was a frequent finding in ATIN (60% of cases), in line with prior findings in smaller studies by Praga et al. (67% of cases) [7] and Fogazzi et al. (48% of cases) [1], but higher than in a larger study (33% of cases) [5]. Together, these and the present study indicate that haematuria is a relatively frequent finding in ATIN. However, no previous study has analysed the significance of haematuria in ATIN. We observed that the presence of haematuria was associated with the presence and magnitude of proteinuria as well as with higher serum creatinine at last follow-up. Moreover, we observed a close relationship between the haematuria and proteinuria, it seems that those patients with more haematuria intensity were those with higher proteinuria levels. In line with these findings, there was a numerically higher need for renal replacement therapy at the end of follow-up among haematuric patients, although the overall number of events was low (*n* = 9) and the difference was not statistically significant (*p* = 0.12). In this regard, a key and serious consequence of ATIN is fibrosis, with a consequent increased risk of developing CKD [19]. Incomplete recovery of renal function has been associated with fibrosis in ATIN patients [15].

Data on the severity of haematuria were available for a subset of patients. These data did not show an association between severity of haematuria and other features of disease severity or renal function outcomes. The limited number of patients with this information makes it difficult to interpret the renal function results.

Detailed histological information was available for a majority of patients. A key finding was the higher frequency of red blood cell casts in patients with haematuria. Red blood cell casts are usually considered to represent glomerular haematuria. Together with the higher proteinuria in patients with haematuria, this may be interpreted as evidence of glomerular involvement in certain cases of ATIN and the association of glomerular involvement with more severe outcomes. While certain causes of ATIN such as nephrotic syndrome induced by NSAIDs have been classically associated with glomerular involvement [20], no association of haematuria with specific causes of ATIN were observed in our study. An alternative explanation for the association of haematuria with proteinuria and ATIN outcomes would be an underlying glomerulopathy. However, cases identified as such by kidney biopsy were carefully excluded before analysis. Finally, both proteinuria and haematuria may cause kidney injury, so mechanistic studies would be required to dissect the specific contribution of each factor or the potential interaction between them to worse outcomes [12,21]. However, several sensitivity analyses failed to find any association between higher levels of proteinuria and worse kidney outcomes. Thus, it appears that haematuria itself, independently of proteinuria action, was associated with a worse renal outcome.

The clinical characteristics of the present series of patients is in line with prior reports. Thus, the severity of AKI was similar to previous studies, with a serum creatinine peak around 5.55 mg/dL [5,15], and proteinuria was below the nephrotic range [5,7]. Moreover, the main cause of ATIN was drug toxicity, in agreement with previous studies in developed countries [22].

Since the main cause of ATIN is drug-induced, rapidly stopping the drug thought to be responsible may be the best therapeutic option. However, this is complicated in patients on multiple medications, explaining why the decision may be delayed, and why 30–70% of patients do not recover their baseline renal function [23,24,25,26,27,28,29]. Due to the key role of the immune system in human ATIN, steroids are often used as treatment but this remains controversial due to the lack of prospective, randomised and controlled clinical trials. Some reviews recommend prescribing steroids in drug-associated ATIN only when renal function does not recover by 7–15 days after drug withdrawal [23]. However, a recent study suggested that an interval greater than 7 days between drug withdrawal and steroid initiation, and the severity of interstitial fibrosis are key factors associated with an increased risk of incomplete recovery of renal function [19]. Steroid treatment has been associated with a better renal function outcome and less need for dialysis [16,30], but these findings have not been universal [6]. In our study, most patients received steroids. However, there was a time lag of several weeks between ATIN onset, as defined by the time at onset serum creatinine assessment and renal biopsy, and steroids were generally initiated after renal biopsy results. No association of steroids with outcomes was observed.

Some limitations of our study should be acknowledged. There was no information on the use of oral anticoagulants. These have been associated with haematuria-induced kidney injury [12,31,32]. However, there were no differences in the prevalence of diseases usually associated with anticoagulant prescription (e.g., ischemic heart disease, stroke or atrial fibrillation) between patients with or without haematuria. Furthermore, data concerning clotting factors and other haemostatic markers, either haemorrhagic or thrombotic, were not available.

Information on the presence or severity of haematuria was absent in a significant percentage of patients, illustrating the attitude of nephrologists towards haematuria in ATIN and further supporting the need to make public the present data in order to encourage further research on the issue. Additionally, baseline haematuria data were not available for most patients. Thus, we cannot definitely exclude that haematuria in some patients may have been associated with a pre-existent condition. However, this does not detract from the observed association of haematuria with outcomes.

## 5. Conclusions

Our results show that haematuria is a frequent finding in patients with ATIN and is associated with parameters of disease severity and worse outcomes. However, it is necessary to validate this observation in further studies involving patients from different backgrounds and ethnicities. Moreover, experimental studies are needed to determine if haematuria is a consequence of the severity of ATIN or a pathogenic factor which aggravates kidney damage and, therefore, might be susceptible to therapeutic intervention.

## Figures and Tables

**Figure 1 jcm-09-02135-f001:**
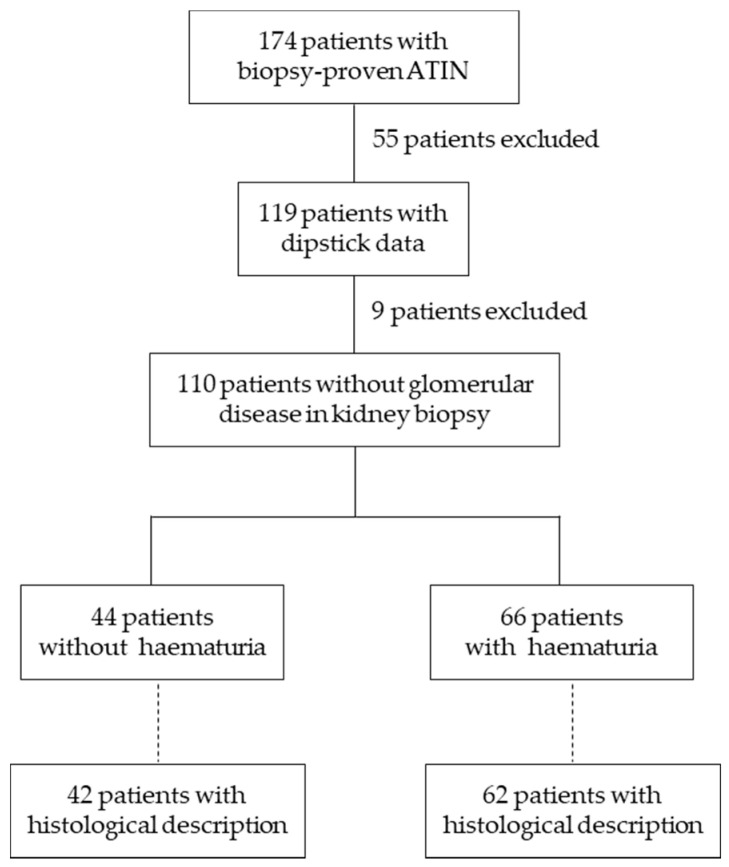
Patient disposal. ATIN, acute tubulointerstitial nephritis.

**Table 1 jcm-09-02135-t001:** Baseline patient characteristics.

Variables	No Haematuria (*n* = 44)	Haematuria (*n* = 66)	*p*-Value
Gender, male (%)	23 (52.3)	36 (54.5)	0.68
Age at diagnosis (years)	60.2 ± 14.1	55.2 ± 18.5	0.14
Cardiovascular disease *n* (%)	23 (52.3)	37 (56.1)	0.37
Hypertension, *n* (%)	14 (31.8)	27 (40.9)	0.19
Diabetes mellitus, *n* (%)	11 (25)	17 (25.7)	0.77
Dyslipidaemia, *n* (%)	0	3 (4.5)	0.14
CKD, *n* (%)	30 (68.2)	41 (62.1)	0.30
Gout, *n* (%)	0	4 (6.1)	0.08
Stroke, *n* (%)	1 (2.3)	3 (4.5)	0.49
Ischemic heart disease, *n* (%)	1 (2.3)	5 (7.6)	0.20
Atrial fibrillation, *n* (%)	3 (6.8)	2 (3)	0.39
Respiratory disease, *n* (%)	4 (9.1)	5 (7.6)	0.85
Asthma, *n* (%)	3 (6.8)	2 (3)	0.39
COPD, *n* (%)	1 (2.3)	1 (1.5)	0.81
Autoimmune disease, *n* (%)	17 (38.6)	16 (24.2)	0.14
Hypothyroidism, *n* (%)	1 (2.3)	4 (6.1)	0.31
Rheumatoid arthritis, *n* (%)	2 (4.5)	1 (1.5)	0.37
Sjogren syndrome, *n* (%)	4 (9.1)	3 (4.5)	0.36
Crohn’s disease, *n* (%)	1 (2.3)	1 (1.5)	0.81
Polymyalgia rheumatica, *n* (%)	2 (4.5)	2 (3)	0.73
Primary biliary cirrhosis, *n* (%)	1 (2.3)	1 (1.5)	0.81
Sarcoidosis, *n* (%)	2 (4.5)	1 (1.5)	0.37
Coeliac disease, *n* (%)	3 (6.8)	1 (1.5)	0.16
Psoriasis, *n* (%)	2 (4.5)	2 (3)	0.73
Haematological disease, *n* (%)	2 (4.5)	4 (6.1)	0.67
MGUS, *n* (%)	2 (4.5)	1 (1.5)	0.37
Infectious disease, *n* (%)	0 (0)	2 (3)	0.23
Tuberculosis, *n* (%)	0 (0)	2 (3)	0.23
Oncological disease, *n* (%)	2 (4.5)	10 (15.1)	0.06
Prostate, *n* (%)	0 (0)	3 (4.5)	0.14
Breast, *n* (%)	2 (4.5)	2 (3)	0.68
Eye disease, *n* (%)	1 (2.3)	4 (6.1)	0.34
Serum Cr (mg/dL)	1.10 ± 0.25	1.21 ± 0.69	0.36
eGFR CKD-EPI (ml/min/1.73 m^2^)	65.06 ± 16.40	66.84 ± 24.77	0.70
UPCR (mg/g)	292 (123–734)	457 (177–887)	0.51

CKD: chronic kidney disease; COPD: chronic obstructive pulmonary disease; eGFR CKD-EPI: estimated glomerular filtration rate using the equation CKD-EPI (Chronic Kidney Disease-Epidemiology Collaboration); MGUS: monoclonal gammopathy of uncertain significance; UPCR: urinary protein:creatinine ratio.

**Table 2 jcm-09-02135-t002:** Acute kidney injury secondary to ATIN according to the presence of haematuria.

Variables	No Haematuria (*n* = 44)	Haematuria (*n* = 66)	*p*-Value
Onset serum Cr (mg/dL)	3.29 ± 2.60	3.91 ± 3.85	0.36
Peak serum Cr (mg/dL)	4.28 ± 3.03	5.55 ± 4.25	0.08
Biopsy serum Cr (mg/dL)	3.17 ± 2.47	3.82 ± 2.66	0.20
Biopsy UPCR (mg/g)	341.00 (177–734)	902.70 (513–1492)	0.001
End of follow-up serum Cr (mg/dL)	1.91 ± 1.17	3.19 ± 2.91	0.007
Other characteristics			
Eosinophilia *n* (%)	12 (27.3)	20 (30.3)	0.77
Eosinophil number (×10^3^/µL)	0.34 ± 0.26	0.34 ± 0.21	0.98
CRP (µg/L)	40.0 ± 82.1	51.8 ± 63.9	0.41
Asymptomatic, *n* (%)	2 (4.5)	2 (3.03)	0.98
Proteinuria, *n* (%)	26 (59.1)	59 (89.4)	0.001
Nephrotic syndrome *n*, (%)	0	2 (2.8)	0.25
Hypertension, *n* (%)	23 (52.3)	35 (53.0)	0.60
AKI, *n* (%)	43 (97.7)	62 (93.9)	0.57
Acute dialysis, *n* (%)	1 (2.3)	7 (10.6)	0.09

UPCR: urinary protein:creatinine ratio; CRP: C reactive protein, AKI: acute kidney injury, ATIN: acute tubulointerstitial nephritis.

**Table 3 jcm-09-02135-t003:** Outcomes according to the presence of haematuria.

Variables	No Haematuria (*n* = 44)	Haematuria (*n* = 66)	*p*-Value
Renal replacement therapy, *n* (%)	1 (2.3)	8 (12.1)	0.12
Recovered kidney function *n* (%)	17 (38.6)	23 (34.8)	0.80
Time to recovery of renal function, days	60 (30–362)	60 (27–120)	0.32
Relapse, *n* (%)	3 (6.8)	4 (6.1)	0.52

**Table 4 jcm-09-02135-t004:** Causes of ATIN according to presence of haematuria.

Variables	No Haematuria (*n* = 44)	Haematuria (*n* = 66)	*p*-Value
1 cause, *n* (%)	26 (59.1)	28 (42.4)	0.15
2 or more causes, *n* (%)	6 (13.6)	9 (13.6)	0.84
Drugs, *n* (%)	27 (61.4)	32 (48.5)	0.34
Antibiotic, *n* (%)	5 (11.4)	6 (9.1)	0.82
NSAID, *n* (%)	2 (4.5)	9 (13.6)	0.08
PPI, *n* (%)	18 (40.9)	17 (25.7)	0.16
ASA, *n* (%)	2 (4.5)	1 (1.5)	0.57
Other drugs, *n* (%)	4 (9.1)	4 (6.1)	0.71
Infection, *n* (%)	1 (2.3)	0	0.42
Systemic disease, *n* (%)	7 (15.9)	9 (13.6)	0.90
Crohn’s, *n* (%)	0	1 (1.5)	1.00
Sjogren, *n* (%)	2 (4.5)	2 (3.0)	1.00
Sarcoidosis, *n* (%)	1 (2.3)	3 (4.5)	0.64
Granulomatous, *n* (%)	2 (4.5)	1 (1.5)	0.57
Other, *n* (%)	2 (4.5)	2 (3.0)	1.00
Unknown, *n* (%)	3 (6.8)	7 (10.6)	0.51
TINU, *n* (%)	0	1 (1.5)	1.00

NSAID: non-steroidal anti-inflammatory drug; PPI: proton pump inhibitor; ASA: aminosalicylic acid (aspirin); TINU: tubulointerstitial nephritis and uveitis.

**Table 5 jcm-09-02135-t005:** Histological findings.

Variables	No Haematuria (*n* = 44)	Haematuria (*n* = 66)	*p*-Value
Interstitial eosinophils, *n* (%)	29 (65.9)	44 (66.7)	0.82
>50% of the tissue, *n* (%)	17 (38.6)	31 (47)	0.30
Neutrophils, *n* (%)	6 (13.6)	13 (19.7)	0.36
Lymphoplasmacytoid cells, *n* (%)	21 (47.7)	35 (53)	0.18
Macrophages, *n* (%)	26 (59.1)	40 (60.6)	0.09
Non-caseating granulomas, *n* (%)	8 (18.2)	17 (25.7)	0.56
Interstitial fibrosis, *n* (%) >50% of the tissue, *n* (%)	20 (45.4) 2 (4.5)	38 (57.6) 4 (6.1)	0.56 1.00
Tubular atrophy, *n* (%) >50%, *n* (%)	28 (63.6) 3 (6.8)	41 (62.1) 8 (12.1)	0.98 0.50
Tubulitis, *n* (%)	24 (54.5)	34 (51.5)	0.10
Hyaline casts, *n* (%)	14 (32.7)	25 (37.5)	0.07
Granular casts, *n* (%)	10 (22.7)	10 (15.1)	0.73
Red blood cell casts, *n* (%)	1 (2.3)	7 (10.6)	0.006
Glomerular sclerosis, *n* (%) >50% of the tissue, *n* (%)	22 (50) 3 (6.8)	32 (48.5) 2 (3)	0.48 0.39
Periglomerular fibrosis, *n* (%)	12 (27.3)	16 (24.2)	1.00
Arteriosclerosis, *n* (%)	25 (56.8)	41 (62.1)	1.00

**Table 6 jcm-09-02135-t006:** Acute kidney injury secondary to ATIN according to haematuria severity.

Variables	Haematuria +++ (*n* = 18)	Haematuria +/++(*n* = 41)	*p*-Value
Onset serum Cr (mg/dL)	3.53 ± 4.45	4.20 ± 3.89	0.57
Peak serum Cr (mg/dL)	5.99 ± 4.64	5.50 ± 4.37	0.69
Biopsy serum Cr (mg/dL)	3.89 ± 2.21	3.91 ± 2.97	0.98
Biopsy UPCR (mg/g)	1352.65 (665–2292)	849.60 (592–1155)	0.02
End of follow-up serum Cr (mg/dL)	3.14 ± 3.11	3.25 ± 2.98	0.90
Characteristics Eosinophilia, *n* (%)	3 (16.6)	15 (36.6)	0.21
Number (×10^3^/µL)	0.26 ± 0.16	0.39 ± 0.23	0.04
CRP (µg/L)	77.51 ± 91.07	41.25 ± 48.91	0.05
Asymptomatic, *n* (%)	1 (5.5)	1 (2.4)	0.53
Proteinuria, *n* (%)	17 (94.4)	37 (90.2)	1.00
Nephrotic syndrome, *n* (%)	1 (5.5)	1 (2.4)	0.53
Hypertension, *n* (%)	11 (61.1)	19 (46.3)	0.36
AKI, *n* (%)	18 (100)	37 (90.2)	0.48
Acute dialysis, *n* (%)	2 (11.1)	5 (12.2)	1.00

UPCR: urinary protein:creatinine ratio; CRP: C reactive protein. +, mild, ++, moderate, +++, severe.

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
