# Peer review of "Hematuria Is Associated with More Severe Acute Tubulointerstitial Nephritis"

_jcm, 2020, doi:10.3390/jcm9072135_

Round 1

Reviewer 1 Report

Acute tubulointerstitial nephritis (ATIN) is a common cause of acute kidney injury. Haematuria is a risk factor for the development of renal disease, however, no study has reported the significance of haematuria in ATIN. The manuscript by Esteras et al conducted a retrospective, observational analysis of 110 patients with ATIN. Their study aimed to determine the prevalence of haematuria in ATIN patients and to report whether haematuria is associated with kidney outcomes. Their study is novel and is a significant contribution to the field, as no previous reports have analyzed the association of haematuria with renal outcomes in ATIN. They found that haematuria is common in patients with ATIN and is associated with worse renal function outcomes.

This work is well organized and comprehensively described. This study is very interesting and offers new insights into the treatment and diagnosis of ATIN.

Author Response

We thank the reviewer for the positive report of our article.

Reviewer 2 Report

JCM_850638 _June2020-30

Dear Authors: Raquel Esteras, Jonathan G Fox, Colin C Geddes, Bruce Mackinnon, Alberto Ortiz and Juan Antonio Moreno

This overview is related to the manuscript entitled “Hematuria is associated with more severe acute tubulointerstitial nephritis” submitted to the Journal of Clinical Medicine (JCM).

This study presents a right approach for retrospective analysis, including its potential bias as pointed in the discussion. The manuscript presents an original text based on an observational study of biopsy-proven ATIN cases between

January 2010 and December 2018. Authors also have used adequate statistic tools to reach reliable results.

Although it is not a mandatory request, the following suggestion may improve the impact of the study. In order to reduce the impact of potential biases and its implications on data interpretation, I recommend that data on clotting factors and other hemostatic markers, both hemorrhagic and thrombotic, could be included to the results and discussion, if it’s possible to be reached on the database.

A high point on this manuscript is the mindful description of the correlation between hematuria and several variables commonly find among AKI patients.

Author Response

We thank the reviewer for the proposal to improve the manuscript. Although we agree with the reviewer that info on these factors would be desirable, unfortunately, this is not available in the database. This is now acknowledged in the limitations section, which has been updated by adding: "However, there were no differences in the prevalence of diseases usually associated with anticoagulant prescription (e.g. ischemic heart disease, stroke or atrial fibrillation) between patients with or without hematuria. Furthermore, data concerning clotting factors and other hemostatic markers, either hemorrhagic or thrombotic, were not available."